evolution, genetics

meiotic drive, *Drosophila pseudoobscura*, polyandry, selfish gene, sperm competition, polymorphism

**Author for correspondence:**
N. Wedell
e-mail: n.wedell@exeter.ac.uk

One contribution to the Special Feature 'Natural and synthetic gene drive systems'. Guest edited by Nina Wedell, Anna Lindholm and Tom Price.

# Ancient gene drives: an evolutionary paradox

T. A. R. Price[1], R. Verspoor[1] and N. Wedell[2]

[1]Institution for Integrative Biology, University of Liverpool, Liverpool L69 7ZB, UK
[2]Biosciences, University of Exeter, Penryn Campus, Penryn TR10 9FE, Cornwall, UK

NW, 0000-0003-1170-8613

Selfish genetic elements such as selfish chromosomes increase their transmission rate relative to the rest of the genome and can generate substantial cost to the organisms that carry them. Such segregation distorters are predicted to either reach fixation (potentially causing population extinction) or, more commonly, promote the evolution of genetic suppression to restore transmission to equality. Many populations show rapid spread of segregation distorters, followed by the rapid evolution of suppression. However, not all drivers display such flux, some instead persisting at stable frequencies in natural populations for decades, perhaps hundreds of thousands of years, with no sign of suppression evolving or the driver spreading to fixation. This represents a major evolutionary paradox. How can drivers be maintained in the long term at stable frequencies? And why has suppression not evolved as in many other gene drive systems? Here, we explore potential factors that may explain the persistence of drive systems, focusing on the ancient sex-ratio driver in the fly *Drosophila pseudoobscura*. We discuss potential solutions to the evolutionary mystery of why suppression does not appear to have evolved in this system, and address how long-term stable frequencies of gene drive can be maintained. Finally, we speculate whether ancient drivers may be functionally and evolutionarily distinct to young drive systems.

## 1. Introduction

Organisms are the product of a network of cooperating genes. This cooperation has evolved because each gene increases its own fitness by making the individual more successful. However, some genes do not cooperate, instead selfishly manipulating the organism to enhance their own success, at a cost to the rest of the genome. Such genes are referred to as selfish genetic elements (SGEs) [1]. Many SGEs manipulate reproduction, enhancing their own transmission to offspring and can spread extremely rapidly through populations, often reaching fixation [1]. However, the costs they impose on the rest of the genome can lead to the evolution of mechanisms that suppress the driving gene, removing its transmission advantage, rendering the gene drive ineffective, typically to be eliminated from the population due to its costs, or simply due to Muller's ratchet [1]. As gene drives are often associated with low recombination, even if a gene drive does persist, it is typically expected to build up deleterious mutations which will gradually reduce its fitness, which again will lead to elimination from the population (e.g. [2]). Hence, gene drives are often expected to be transient, either spreading rapidly to fixation or being eliminated by suppression (e.g. [3,4]). However, there are some gene drive systems that seem to have avoided these two fates, instead being maintained at stable frequencies in natural populations, potentially for hundreds of thousands of years [5]. This represents a major evolutionary paradox. How can they be maintained in the long term at stable frequencies, without either degrading by failing to maintain fitness or evolving stronger drive that allows them to reach fixation? And, more puzzling, why has suppression not evolved to counter their

mechanisms of drive and eliminate them entirely as is the case in most other gene drive systems?

Here, we focus on one of the best-studied classes of SGEs, the meiotic driving segregation distorters [5]. These selfish chromosomes manipulate gametogenesis to ensure that they are passed on to more than the fair Mendelian ratio of 50% of viable gametes [6]. The fate of gene drive systems in a population is typically expected to follow one of two courses: either its transmission advantage allows it to spread rapidly to fixation or it is subject to counter-evolution to suppress drive that renders the drive system ineffective [1]. Avoiding the first fate (suppression) is critical when deploying man-made gene drive systems and, depending on the strategy adopted, achieving the second (fixation) may also be desirable. Empirical and theoretical work with CRISPR/Cas9-based synthetic gene drives supports the prediction that resistance rapidly evolves, with suppression occurring in as little as one generation [7,8]. There is also substantial evidence of suppression from natural gene drive systems, with several gene drives being completely suppressed. If genetic suppression does not evolve rapidly enough, theory typically predicts that a strong driver will rapidly spread to fixation [9]. If the driver occurs on a sex chromosome, this will eliminate the driver and the population. If the driver is autosomal, it will reach fixation, and all individuals will be homozygous for drive. The *Paris* drive system in *Drosophila simulans* is a stunning example of the predicted dynamics of an X-chromosome driver [10,11]. Over the past 2 decades, the *Paris* driver spread rapidly out from southeast Africa, spreading west and invading *D. simulans* populations in the Indian ocean and north through east Africa up to Egypt. When *Paris* reached populations, it rapidly attained high frequencies, creating strongly female-biased populations. However, the spread of the driver was followed by the rapid spread of alleles on the autosomes and Y chromosome that suppress the drive mechanism. These suppressors lagged behind the expansion of *Paris* by only a matter of years. When the suppressors reached a population, they spread rapidly, rendering *Paris* unable to drive in that population, and normalizing the population sex ratio. This expanding chase of driver and suppressor continues to this day, spreading across North Africa and into Asia and Europe [10,11]. This system illustrates how rapidly gene drives can spread geographically and within populations, and how rapidly suppression can disable a drive system.

However, not all meiotic drive systems follow this dynamic pattern. Instead a surprising proportion of well-known X-chromosome meiotic drive systems are found at apparently stable equilibria in natural populations, often showing geographical clines in frequency (table 1). In some cases, these frequency clines have existed, apparently unchanged, for decades. All these drive systems create female-biased offspring sex ratios, by killing sperm and probably reducing the fertilization success of males that carry the driver. So, they impose substantial costs on the rest of the genome, which should strongly select for the evolution of suppression, or of increasingly effective suppression [1]. So, why do we not observe suppression in these drive systems? And if they are not suppressed, why do they not spread rapidly to fixation? Theory suggests that drivers should be under strong selection to improve their strength of drive, with stronger drivers outcompeting weaker variants [4]. The evolution of even a small advantage in drive, or a

reduction in costs of drive, should throw the system off balance and allow the driver to reach fixation. And yet we do not see this in nature. Moreover, the autosomal *t*-haplotype system in mice, and the sex-ratio distorting SR X-chromosome drive system in *D. pseudoobscura* that appear to be ancient, even hundreds of thousands to millions of years old, show no evidence of suppression of their drive mechanisms [20]. In *D. pseudoobscura*, SR display the same frequency cline across the USA today as Theodosius Dobzhansky documented when studying the system in the 1930s–1950s [12]. Yet theory suggests that drive in natural populations should be evolutionarily unstable: either leading to rapid fixation, or the driver is doomed to the inevitable evolution of suppression, loss of effective drive and the elimination of the drive system [4,9,21,22]. The observed long-term maintenance of balanced SR polymorphism in *D. pseudoobscura* is therefore unexpected.

So, these ancient drive systems pose two great mysteries. Why are they not suppressed? And why do they persist at stable frequencies in natural populations? Regarding the question of evolution of suppression, we suggest five possibilities. First, we are wrong: there is suppression, and we just have not detected it. Second, a lack of suppression could occur because the costs of drive are low, and that the rest of the genome has instead evolved to tolerate and reduce the costs of drive. Third, there could be something unusual about these species, making it particularly difficult for them to evolve suppression of drive. A fourth possibility is that suppression should evolve, but that by chance, the mutations that would lead to suppression have not yet occurred. Finally, there is the possibility that some drive systems become impossible, or almost impossible, to suppress.

Here, we focus on X-chromosome meiotic drive in the fruit fly *Drosophila pseudoobscura* as a case study to illustrate the issues surrounding the persistence of an ancient unsuppressed gene driver and to highlight the mechanisms that may maintain long-term stable polymorphisms in nature. We discuss potential solutions to the evolutionary mystery of why suppression does not appear to have evolved. We also address how long-term polymorphism can be maintained together with the potential for rapid changes in the transmission and cost of the gene driver. While the molecular basis to SR drive is not yet known in *D. pseudoobscura*, the ecological genetics, including the frequency and consequences of SR, has been extensively examined since the 1940s (e.g. [13]), making it one of the best-studied meiotic drive systems in the laboratory and the wild.

## 2. The sex-ratio paradox I: an ancient gene drive system with no signs of suppression

*Drosophila pseudoobscura* is an extremely common fly found in the woodlands of western North America. It occurs widely from the temperate rainforests of Canada to the mountain forests of southern Mexico and Guatemala, with an isolated population persisting in Colombia [13]. The best estimates of population size suggest there is one *D. pseudoobscura* for every 10 m$^2$ of forest in the western USA, and population genetics estimate typical population sizes in the hundreds of thousands [23,24]. As well as being abundant, *D. pseudoobscura* is a capable flyer and can disperse easily between habitat patches. The combination of substantial population

**Table 1.** Examples of five *Drosophila* species that show geographical distribution in the frequency of their X-chromosome drive systems. The duration column shows the duration between the first observation of the distribution to the most recent work verifying that the distribution has not changed, for species where multiple surveys have been published.

| species | pattern of drive distribution | duration distribution has been observed | references |
|---|---|---|---|
| *D. pseudoobscura* | clinal, drive absent in north, evidence of decline in far south | 1930s–2014 | [12–14] |
| *D. neotestacea* | clinal, drive rare in north | 1990–2013 | [15,16] |
| *D. persimilis* | clinal, drive absent in north, very limited evidence of decline in far south | n.a. | [13] |
| *D. subobscura* | drive limited to North Africa, absent in Europe | 1960s–2015 | [17,18] |
| *D. paramelanica* | rare in north, less rare in south | n.a. | [19] |

size and dispersal ability is probably responsible for the very high rates of gene flow seen across North America [25].

SR drive is found in many populations of *D. pseudoobscura* in Mexico and the USA. It occurs at a maximum frequency of approximately 30% in populations on the USA/Mexico border, with frequencies decreasing in a latitudinal cline towards Canada, where it is absent [12,13,26] (figure 1). South into Mexico, its frequencies decline towards Guatemala [14]. The SR cline in the USA seems to have been stable for at least the past 80 years [12]. The phenotype, strength of drive and karyotype have remained consistent, suggesting it has been the same SR chromosome throughout this period. Males who carry the SR chromosome transmit it to all their offspring, producing only daughters, never fertile sons. SR gene drive exists within a unique set of chromosomal inversions on the right arm of the X chromosome, that is both ancient in origin (more than 1 Myr old) and highly diverged compared to standard (ST) X-chromosome arrangements [28]. SR causes the failure of Y-chromatid segregation at meiosis II, with downstream developmental problems occurring during sperm elongation and individualization [29] that essentially kills all Y-bearing sperm. As yet, the mechanisms of how Y-sperm are being targeted and killed remain unknown. Despite SR being both a very strong driver and being very old, no evidence of genetic suppression in the system has been found to date.

Is it possible that suppression currently exists but has not yet been detected? We argue that this is unlikely for three reasons. First, SR is by far one of the best and most extensively studied drive systems in the field. There have been multiple large-scale studies that should have found suppression if it exists, in the 1940s [13], 1970s [30], 1980s [31,32] and 2010s [12]. So if suppressors of SR exist, they would have been detected. Second, if suppression has evolved, it should be observed in or close to populations harbouring drive, which is exactly the type of populations where researchers have looked the hardest for signals of suppression. Third, we would expect suppression, unless enormously costly, to spread to all populations where SR is present, as has been seen in other similar systems in related *Drosophila* species (e.g. *D. simulans* [10]). Another possibility is that coevolution between drive and suppressors has been cycling, with novel suppressors emerging, suppressing SR and novel SR variants escaping suppression. However, the lack of change in the karyotype and phenotype, as well as no evidence of a mosaic of suppression and effective/ineffective SR variants, makes this unlikely. Moreover, any selection powerful enough to replace

an SR variant with a fitter one is likely to also alter the overall fitness of SR in nature. Yet, the frequency of SR in populations has remained remarkably stable over time [12]. We therefore conclude that the lack of suppression in this system is real and is not simply due to field sampling having overlooked it.

How costly a driver is will affect the strength of selection for suppression to evolve. One argument could therefore be that SR is simply not costly enough to trigger the evolution of suppression. This is unlikely, as the SR distortion of broods is likely to impose substantial costs in nature. As populations become increasingly female-skewed due to the presence of SR, the ability of the genome to suppress this bias and produce the rare sex comes under increasingly strong selection. In addition, the *D. pseudoobscura* SR chromosome, due to its reduced effective population size and limited recombination that spans over two-thirds of the X chromosome, provides a large target for the accumulation of deleterious mutations [33]. Moreover, experimental work has also shown that SR has the potential to rapidly sweep through populations, bias sex ratios and cause population extinction [34]. So, the long-term stable frequencies of SR in nature are a major mismatch between laboratory studies, theory and field observations.

Is it possible there is a species-specific factor that makes it more difficult for suppression to evolve in *D. pseudoobscura* compared to other *Drosophila* species? For example, *D. pseudoobscura* might have reduced RNAi silencing efficiency, as RNAi has been shown to be important for suppression in several gene drive systems [35,36], or lost some other defensive system. However, the young *Overdrive* system found in the isolated *D. pseudoobscura bogotana* subspecies shows complete genetic suppression in its host subspecies [37]. Indeed, the system was only discovered when hybrid 'sterile' F1 male crosses between the subspecies produced small amounts of viable sperm that, when aged, resulted in female-biased offspring [37,38]. So, clearly *D. pseudoobscura* as a species can evolve suppression, albeit to a different and younger meiotic driver. Alternatively, it could be simply chance that the mutations that would allow suppression of SR have not occurred in *D. pseudoobscura* US populations. However as mentioned, the SR driver is ancient, so there has been ample time for suppression to evolve [39]. Another possibility is that there may be coevolutionary cycles of drive and suppression, and that a novel form of SR has recently arisen, escaped suppression and replaced older forms of SR across the species' range. However, this possibility also does

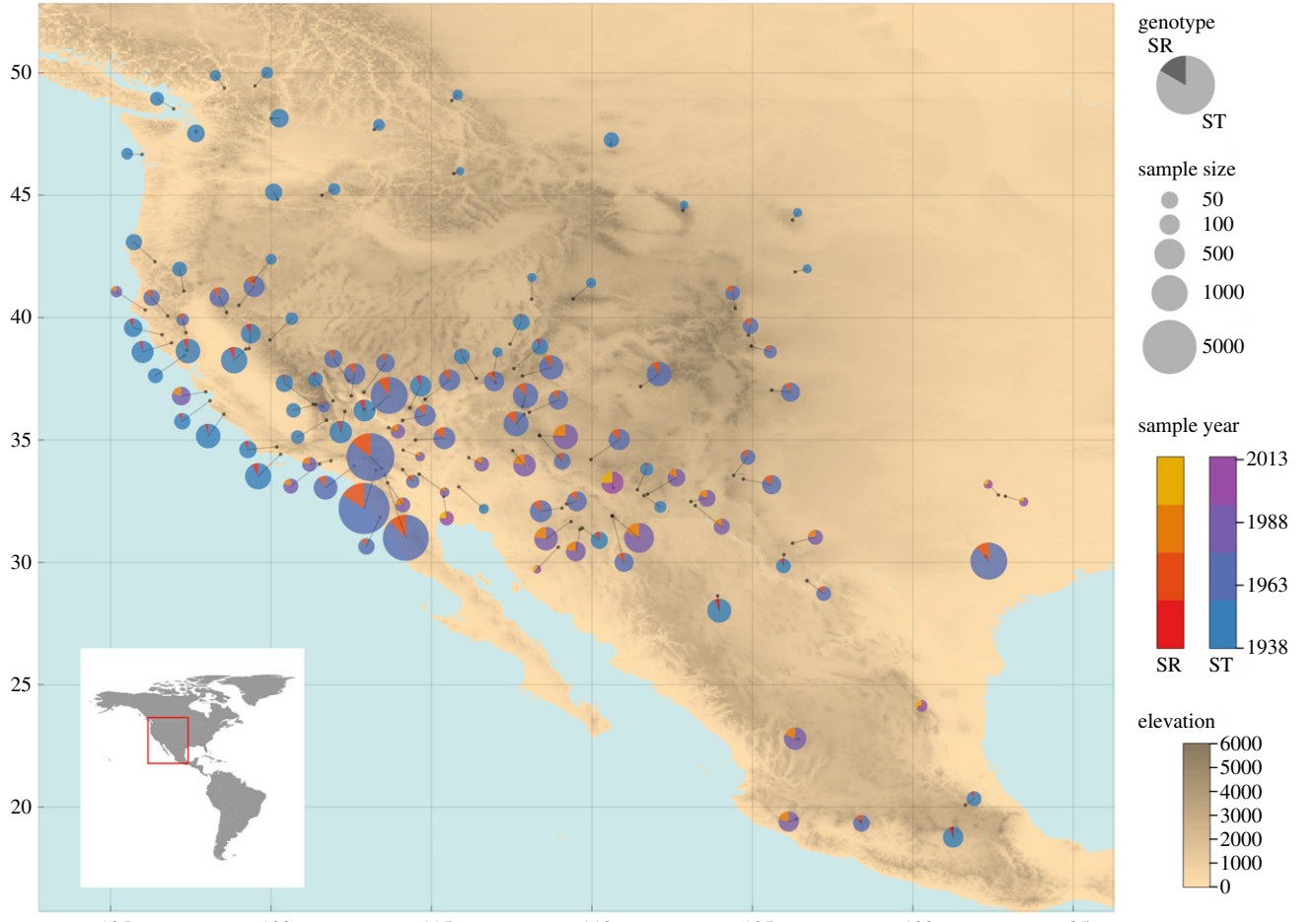

**Figure 1.** A map showing the distribution of SR in *D. pseudoobscura* flies collected between 1938 and 2013 across western North and Central America. This map was made using the publicly available elevation data: CIAT-CSI SRTM (http://srtm.csi.cgiar.org) [27]. (Online version in colour.)

not seem likely. SR has been unsuppressed for at least the past 80 years, and the high costs of SR in terms of sex ratio bias, the large population sizes, high gene flow and many generations per year of *D. pseudoobscura*, suggest there should be ample opportunity for suppression to evolve since SR was first discovered in the 1930s [13]. In addition, *D. pseudoobscura* is not the only species that harbours a stable and old unsuppressed drive system (table 1), suggesting this unusual lack of suppression may extend also to other gene drive systems. It seems unlikely therefore that something specific to only *D. pseudoobscura* explains the lack of suppressions.

Host genomes are known to rapidly evolve suppression to many young drive systems, including the *Overdrive* system in *D. pseudoobscura*. So, is there something about the evolution of SR, over its million-year history, that makes it difficult to suppress? Apart from spore killers where single loci can confer both drive and resistance [40], in most organisms, the simplest case of SR requires two loci; a drive locus that attacks a sensitivity locus in the non-driving ST X chromosome. However, as mentioned, SR is both ancient and is contained inside a large non-recombining region on the X-chromosome, which indicates it is a far more complex system than the simple two-locus model commonly required for drive. The inversion system may provide scope for the accumulations of alleles and mutations that have co-evolved with each other to shape a complex drive system [28]. Consider the possibility that in the million years since SR first appeared, it has been co-evolving with suppressors in a genetic arms race and accumulating enhancers of distortion or

silencers of suppressors within the inversion complex [28]. If the driver experiences long periods when suppressors have successfully been rendered ineffective, and in turn those suppressors are costly, we expect suppressors to be lost from the population. Such a situation could allow the genetic machinery of SR to dramatically increase in complexity [28] and one can imagine that to re-evolve effective suppression would require a far more complex system of suppressors, and thus be much less likely to evolve (figure 2). While the genes that cause drive remain elusive in the SR system, it seems feasible, or even likely, that several loci are involved in maintaining SR gene drive in *D. pseudoobscura*. Could it be that of the more than 2100 genes locked in the SR inversions, those that cause drive itself pose an increasingly insurmountable challenge for the evolution of suppression as time goes on? As SR is not the only ancient gene drive system, it is possible or even likely that a number of other unsuppressed drive systems could also perhaps harbour a complex suite of enhancers of drive through linkage of large chromosomal regions? This suggestion is yet to be determined.

However, this scenario suggests that SR may bear a suite of modifiers of drive that evolved to allow SR to evade suppressors, but that no longer exist. This suggestion raises the question of why these modifiers do not degrade over time. Moreover, if these modifiers have costs, but no immediate function as their suppressors no longer exist, why is SR not replaced by variants of SR that evolve to lose these modifiers? Ultimately, we need to understand the genetic machinery

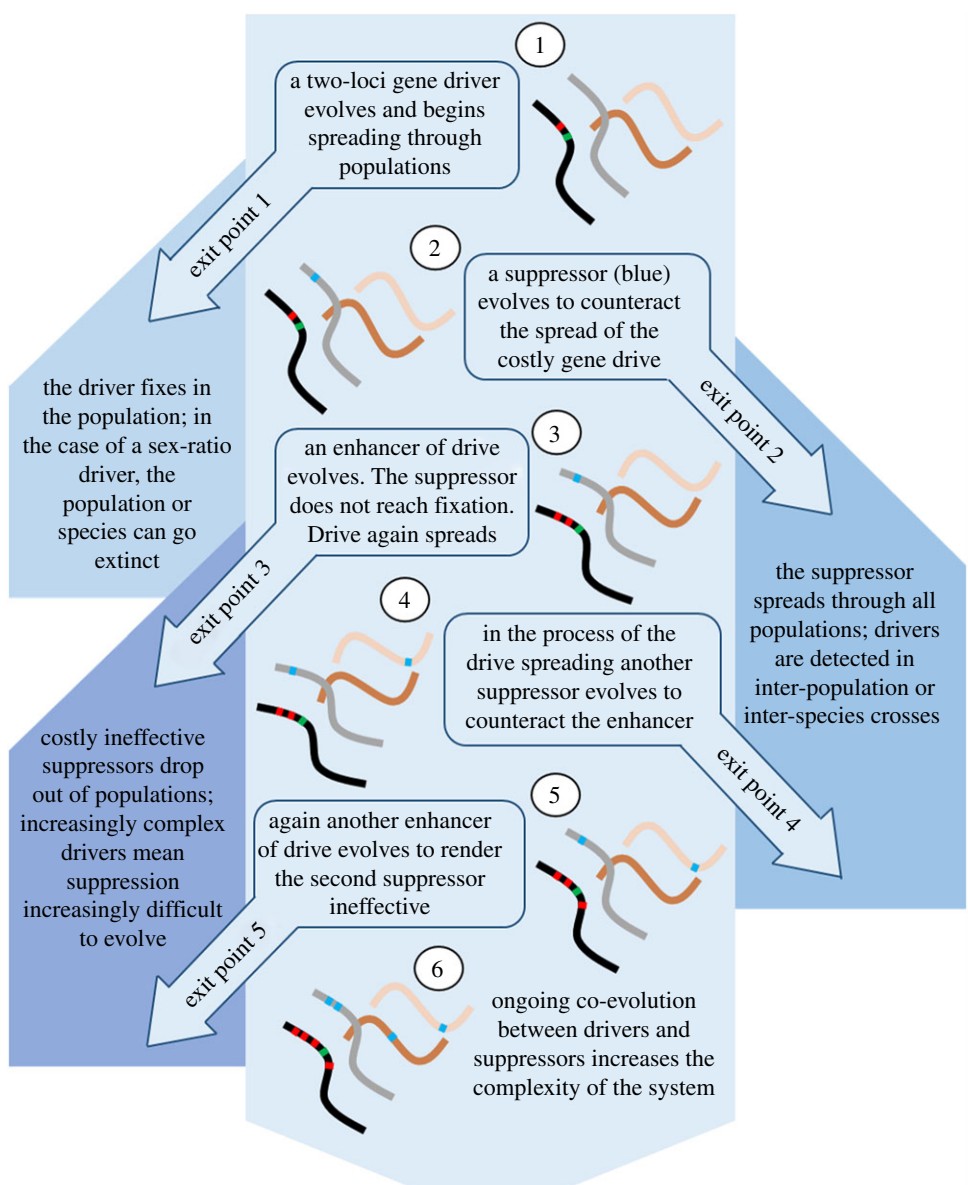

**Figure 2.** An illustration of how ongoing conflict between a driver and its modifiers versus suppressors may create complexity in the system. The drive chromosome (black) begins with a simple two-locus drive system, a killer locus linked to an insensitive responder (step 1). The non-driving homologous chromosome or non-homologous chromosomes can respond by evolving a suppressor (step 2). This suppression can in turn be counteracted by further enhancers of drive (step 3). Cyclical counter-evolution can increase the system's complexity (steps 2–6). At each step, the system can potentially break down either leaving gene drive completely suppressed (exit points 2 and 4) or extremely difficult to suppress (exit points 3 and 5). (Online version in colour.)

underlying why SR drive in *D. pseudoobscura* appears to lack genetic suppression.

## 3. The sex-ratio paradox II: the stability of strong gene drive across a species range

The lack of suppression is only one of the unexpected aspects of the SR system in *D. pseudoobscura*. When a gene drive has a complete transmission advantage, and there is no suppression of the drive mechanism, the driver should spread rapidly within and between populations, unless the driver has major fitness costs that hold it in check. Why then does SR persist at stable intermediate frequencies across multiple populations without reaching fixation? The complex, apparently stable SR frequencies in North America suggest strong forces must be counteracting the selfish transmission advantage of SR. An additional layer of complexity in the SR

system stems from the fact that drive frequency varies in a clinal fashion (figure 1). This pattern can only be explained if there are factors that balance the transmission advantage of SR within a given population. Moreover, some of these factors must also vary between populations, creating the observed cline, as there is evidence of gene flow. Here, we discuss six factors that could play a role in explaining the frequency patterns of SR observed in nature: the shortage of males and male fertility costs, female choice, polyandry and sperm competition, cost to females, population structure, and meta-population dynamics.

## 4. Shortage of males and costs to males through reduced fertility

An obvious cost of harbouring an effective SR distorter is that it can generate female-biased population sex ratios [41]. This

in turn may reduce female fecundity due to lack of sufficient sperm to fertilize all their eggs. Moreover, SR drives by eliminating all non-carrying Y chromosome spermatozoa [42], which means that SR males produce fewer sperm compared with normal males [43]. There is, however, no indication that females suffer sperm limitation due to a female-biased sex ratio even in populations harbouring high frequency of SR and while SR males do produce less sperm than normal males, they still provide sufficient sperm to ensure high fertility in females [43]. Males are also able to mate to multiple females in quick succession [44]. SR males do suffer further reductions in fertility when they experience high temperatures, but this is unlikely to be important in nature, because SR is most common in high-temperature areas [45]. Taken together, this information suggests that even in highly female-biased SR populations, it is unlikely that female fertility will be reduced due to lack of access to sperm.

## 5. Mate choice

If SR males were strongly discriminated against by females, SR males would have reduced mating success meaning that female mate choice is analogous to a genetic suppressor of drive [4,9]. As avoiding mating with SR males will ensure females increasingly produce rare males as offspring, mate choice against SR could feasibly come under selection [46]. Across gene drive systems, there appears to be only limited support for pre-copulatory mate choice against males carrying SR distorters [47–49]. In the *D. pseudoobscura* SR system, there is no evidence that females discriminate between SR and ST males [46].

## 6. Polyandry

Female *D. pseudoobscura* generally mate with multiple males in the wild [43,50,51], which results in sperm competition. Generally, males that produce more sperm enjoy increased paternity success [52]. However, SR males are poor sperm competitors due to producing and transferring fewer sperm at mating [43]. Similar patterns are found in a variety of taxa, where males carrying drivers that kill sperm during meiosis have poor sperm competitive abilities [47,53,54], although this pattern is not universal [55,56]. A consequence of the poor sperm competitive ability of SR males is that their relative fitness is lower in populations with higher frequency of polyandry [57,58]. Experimental evidence demonstrate that the presence of SR can favour increased frequencies of female remating; females evolving in populations with SR males rapidly evolved increased level of polyandry [58]. Polyandry in turn can undermine the transmission advantage of SR in laboratory populations, thus reducing the likelihood that they will go extinct [34]. Recent modelling work also shows that coevolution of female polyandry and meiotic drive can reduce the population frequency of drive [59,60]. Multiple mating is therefore an effective strategy by females to bias paternity against SR males, and as a consequence, polyandry undermines the transmission advantage of SR.

Having demonstrated that SR is disadvantaged by polyandry, it is worth exploring if this behaviour could also contribute to between-population differences in SR frequency. Remarkably, a field study of female remating frequency across populations found the level of polyandry varied considerably. The level of polyandry also exhibits a latitudinal cline across the USA that covaries with the stable cline in SR [12]. Specifically, in northern populations, females have high remating frequencies and SR frequency is low, whereas in southern populations, the reverse is true [12]. This pattern suggests that female multiple mating can effectively reduce the frequency of SR in *D. pseudoobscura* and may be analogous to a genetic suppressor of drive. Interestingly, similar patterns have also been uncovered in another species; *D. neotestacea* shows a cline across North America, and this cline covaries with the frequency of polyandry [15]. However, it is not clear why this cline in polyandry across the USA exists. Polyandry in *D. pseudoobscura* is largely under female control, with males having little ability to suppress female remating [34]. One possibility is that females become more polyandrous if reared or mating at lower temperatures, as the northern higher polyandry populations are likely to experience cooler conditions. However, laboratory work finds no effect of temperature on the level of polyandry [61]. Moreover, the cline in polyandry is at least partly genetically determined [61,62] and the level of polyandry can evolve rapidly in laboratory populations [58]. This finding suggests that there are factors selecting for higher level of polyandry in the northern populations where SR is rare, and conversely for lower polyandry levels in the southern populations. If this model is correct, selection for high and low polyandry across the cline is occurring, and the resulting population level of polyandry determines the transmission success and therefore frequency of SR. Why polyandry would be selected for at higher and lower frequency across the USA is currently unknown. Nevertheless, current evidence suggests that the cline in polyandry will have a major impact on the success of SR males, and this may be largely responsible for the observed cline in SR. However, a recent model based on the SR system suggests that the polyandry cline alone is not enough to create a stable polymorphism for SR in natural populations [59]. Instead, the model suggests additional costs are required to stabilize the frequency of SR [60] (table 2).

## 7. Cost to females

Theory predicts that if a meiotic drive chromosome has high costs when homozygous, this can result in a stable equilibrium of SR and ST chromosomes as it creates negative frequency-dependent selection against the drive allele [59,64]. In other words, as the driver increases in frequency in the population, drive chromosomes will increasingly be found in homozygotes, to the point where the homozygous costs counteract the transmission advantage of the driver [50,59,65,66]. In an X-chromosome drive system, homozygous costs are only expressed in females. Theory also suggests drive systems associated with inversions, such as SR, may be likely to accumulate additional costs [65]. As mentioned, the SR driver occurs in a non-recombining X-chromosome inversion system, which harbours greater than 2100 genes in linkage disequilibrium, representing a large mutational target facilitating the accumulation of deleterious alleles [28]. In addition, the SR X chromosome in *D. pseudoobscura* occurs at low to moderate frequencies (approx. 1–30% [12,67]), meaning that it also has a low effective population size [13]. This will reduce the efficacy of

**Table 2.** Summary of factors in *D. pseudoobscura* that may promote a balanced polymorphism of SR and ST chromosomes.

| factor | evidence | references |
|---|---|---|
| shortage of males | weak | [43,58] |
| reduced male fertility | mixed | [42,43,45,46] |
| mate choice | strong against | [45,46] |
| polyandry and sexual selection | strong for | [12,34,43,59] |
| cost to females | moderate for | [32,60,63] |
| population structure | insufficient data | |

selection on competing driving X haplotypes, allowing more mutations to accumulate. In support of this suggestion is that other drivers in *D. melanogaster* and *D. recens*, for example, are known to result in reduced homozygote fitness [2,68].

However, the search for homozygous costs of SR in *D. pseudoobscura* has produced mixed and even contradictory results. Wallace [63] and Beckenbach [32] found that homozygous SR females had reduced fecundity compared to heterozygous SR/ST females and non-carrying ST females. Wallace also found that heterozygote SR/ST females had substantially higher fecundity than homozygous ST females. A study by Curtsinger & Feldman [65] found little evidence of fecundity costs to homozygous SR females, instead finding that SR/ST heterozygotes had the highest fecundity. Powell [69] also examined this hypothesis and found no difference in fitness between SR and ST females, but concluded that this result may be confounded by examining flies with different chromosome arrangement on the third chromosome. At closer inspection, it seems the differences between the studies may in part be explained by methodological differences, as fecundity was pooled across females [63], and that there may be an impact of larval density [70]. All studies, however, found a cost to carrying SR. We have recently examined the potential fitness cost to homozygous SR females by comparing the impact of carrying a non-driving, or one, or two SR X chromosomes in the same genetic background. We showed that homozygous SR females produce fewer than half as many offspring as heterozygous and homozygous ST females while controlling for genetic background, whereas there was no significant difference in offspring production between heterozygote and homozygote ST females [60]. This substantive fitness cost to homozygous SR females together with polyandry will contribute to generating balancing selection required to maintain SR at a polymorphic frequency as predicted [59,60].

## 8. Population structure and meta-population dynamics

Most natural populations show some form of population structure (viscosity) meaning that some individuals are more likely to interact than others [71]. Many populations are also characterized by localized extinction and recolonization events in a meta-population network [72]. Such meta-population structure can have direct impact on the dynamics of SR distorters by creating heterogeneity in frequency of SR carriers, productivity (offspring production) and probability of extinction due to lack of one sex [3]. It is suggested population structure, through localized extinction and recolonization events in subpopulations, has the capacity to maintain drivers such as SR across wider areas and prevent it from reaching fixation at the species level [4]. This means that variation in productivity between subpopulations coupled with dispersal rates of SR and ST individuals may mean there will be local and even short-term equilibria of SR frequencies in the meta-population network, rather than an overall 'global' equilibrium. The observation of large differences between *D. pseudoobscura* populations in SR frequency that have remained stable for greater than 80 years may suggest the existence of such population structure affecting the frequency of SR. However, these differences remain despite evidence of extensive gene flow between populations [25], indicating that there is ongoing selection within populations that maintain these stable frequencies. To date, there is insufficient information to evaluate the role of population structure for maintaining variable frequencies of SR in natural *D. pseudoobscura* populations, and even the extent of population viscosity despite evidence of ongoing gene flow. However, it is likely that local differences between *D. pseudoobscura* populations across North America will contribute to population structure.

## 9. The stable polymorphism of SR frequencies in populations

For a stable polymorphism between SR and ST to persist, there must be balancing selection to maintain it. Several theories about the factors that prevent SR spreading have not been supported by empirical tests (table 2). However, there seems to be reasonable evidence that polyandry can reduce the fitness of SR in natural populations [12], and that SR imposes costs on females [32,60]. There is good theory suggesting that a combination of the observed cline in polyandry plus high fitness costs to homozygous SR females could lead to a stable cline in SR across the USA [59]. So, we have a reasonable theory about how the cline is currently stabilized. There is some evidence for stability and/or clines in the frequencies of meiotic driving X chromosomes in many fly species (table 1). In particular, work by Kelly Dyer's laboratory on *D. neotestacea* has found good evidence both of a similar north/south frequency cline in meiotic drive, with drive being rarer in the north that parallels an opposite cline in polyandry, similar to the situation in *D. pseudoobscura* [15]. So, it is possible that polyandry is a major determinant of drive frequency in multiple sperm-killing X-chromosome drive systems.

However, even if we can explain the current factors that maintain SR drive in populations at a particular frequency, this is still an unsatisfactory general explanation. Regarding SR in *D. pseudoobscura*, it is not just that SR is stable in populations along a cline, what is remarkable is that it has been at stable frequencies for more than 80 years, perhaps for greater than 500 generations. Again, the question is not just why this cline is stable in the short term, but why it is stable in the long term? If SR has low effective population size and reduced recombination, why does its fitness not decrease over time? Or to ask the same question another way, why does the ST X chromosome not evolve increased fitness allowing it

to outcompete SR across its geographical range? SR imposes substantial costs on females that mate with male bearers, so why do female not evolve ways to detect and avoid SR mates?

## 10. Closing remarks

In this review, we have argued that ancient drive systems such as SR in *D. pseudoobscura* and the *t*-haplotype in mice may be functionally and evolutionarily distinct to young drive systems. But is this true? Are ancient drive systems fundamentally different to more recent ones? Or are they simply part of a continuum; drive systems that just happen to have persisted for longer? Or is there a process that allows a drive system to remain successful long term by escaping the likelihood of direct suppression? Are there mechanisms of drive that are almost impossible to suppress? Are ancient drive systems that combine a near-unsuppressable drive mechanism with sufficient costs of the driver sufficiently complex to prevent fixation? Currently, we do not have enough information to answer these questions. However, there are some potentially fruitful approaches that can be adopted to address this conundrum. First, we need to increase the number of drive systems known. Ideally, we want an unbiased sample of meiotic drive systems rather than the current suite of known systems that is probably heavily biased towards genetic model systems (e.g. mice and flies), and strong SR distorters that create distinctive sex ratios in offspring broods and therefore more easy to detect. Are there large numbers of weak drivers present, or do drive systems rapidly evolve to high transmission advantage? A second key approach is to unravel the genetic mechanism of drive in more systems. While we may have a good understanding of the molecular basis to drive in some plants (e.g. [73]) and fungi (e.g. [40]), in animals currently only drive systems in *D. melanogaster*, mice and perhaps *D. simulans* can be considered well understood, and even they are likely to reveal additional complexities. Although the complexities of the *t*-complex drive system are consistent with the idea that ancient drives are distinct, we simply do not have sufficient additional examples of persistent meiotic drivers to contrast it with. Finally, to date, evolutionary models of meiotic drive systems predict they are unlikely to be stable over evolutionary time. However, the current models do not predict the existence of costly drive systems to persist for millions of years without the evolution of effective suppression, but drivers are nevertheless found at equilibrium frequencies in natural populations that have lasted at least for decades. Perhaps, a new suite of meiotic drive models can propose an explanation for this evolutionary mystery. What is clear is that meiotic drive systems are more dynamic and complex than initially predicted, and that unravelling the factors that maintain them at equilibrium in natural populations for long periods of time has the potential to provide key insight in how to design long-lasting and unsuppressable synthetic gene drivers.

Data accessibility. This article has no additional data.

Competing interests. We declare we have no competing interests.

Funding. T.A.R.P. and R.V. were funded by the Natural Environment Research Council (grant no. NE/S001050/1) and N.W. by grant no. NE/I027711/1.

Acknowledgements. The authors thank Dr Andri Manser for creating figure 1.

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
