## [Reviewer comments · Proceedings of the Royal Society B: Biological Sciences]

Review History

RSPB-2019-2267.R0 (Original submission)

Review form: Reviewer 1

Recommendation

Accept with minor revision (please list in comments)

Scientific importance: Is the manuscript an original and important contribution to its field?

Excellent

General interest: Is the paper of sufficient general interest?

Good

Quality of the paper: Is the overall quality of the paper suitable?

Good

Is the length of the paper justified?

Yes

Should the paper be seen by a specialist statistical reviewer?

No

Do you have any concerns about statistical analyses in this paper? If so, please specify them explicitly in your report.

No

It is a condition of publication that authors make their supporting data, code and materials available - either as supplementary material or hosted in an external repository. Please rate, if applicable, the supporting data on the following criteria.

Is it accessible?

N/A

Is it clear?

N/A

Is it adequate?

N/A

Do you have any ethical concerns with this paper?

No

Comments to the Author

Price, Verspoor and Wedell present a thought-provoking discussion of factors that could lead to long-term persistence of meiotic drivers. I thank the authors for highlighting such an interesting but understudied area of the meiotic drive field. I hope this work stimulates more analysis in this area which may further challenge the assumption that meiotic drivers are generally short-lived. I think the discussion in this work is sound, but I offer a few suggestions below intended to be helpful in improving the paper.

Major Comment:

Using the *D. pseudoobscura* SR system as the primary example is challenging because the driver(s) responsible for the phenotype is (are) unmapped. I think focusing on SR is fine, but I suggest that the authors lay out explicitly that it is unmapped when SR is introduced. In addition, the main arguments are based on the assumption that one genetic drive system underlies the observed SR phenotypes studied since the 1930s. I found it difficult to shake the possibility that there were multiple drivers while reading the paper because several species harbor multiple meiotic drive systems. The conflicting results regarding female fecundity could also be explained by multiple drivers. It would therefore be useful for the authors to lay out their arguments for why they assume that one driver, rather than several, underlies all the observations over time. For example, the authors cite a paper from 1965 regarding the mechanism of SR. Does the phenotype look similar in SR flies collected more recently? Importantly, I am not suggesting that a reader needs to be absolutely convinced about one driver underlying SR for the rest of the discussion to be stimulating/informative. I do appreciate that the authors are using SR as a model to illustrate ideas that could apply broadly to generic drive systems.

Minor Comments:

-Table 1: Why is *D. paramelanica* on this table? There is no range in dates provided. For *D. neotestacea*, why does the range go to 2014 when the last paper was published in 2013?

-p 10: The simplest drive systems do not require 2 loci. The Spok and wtf systems in fungi both work with one gene.

-Figure 1: Distinguishing the colors within the orange and blue groups is difficult. Perhaps use a more dramatically different color palette.

-Figure 2: The placement of the suppressors might give the impression that they must be at an allelic locus (e.g. on the Y chromosome). That is not true and would really limit the genetic real estate available to suppressors. That could lead a reader to be less impressed than they should be at the lack of observed suppressors. Then again, did any of the searches for suppressors look for autosomal suppressors of SR, or only resistant Y chromosomes?

-p 14: Is the Brandt reference a typo?

-p 17: I disagree that those are the only 'well understood' drive systems. There are systems in plants and fungi that are at least as well (sometimes better) understood than those in animals.

Review form: Reviewer 2

Recommendation

Accept with minor revision (please list in comments)

Scientific importance: Is the manuscript an original and important contribution to its field?

Excellent

General interest: Is the paper of sufficient general interest?

Excellent

Quality of the paper: Is the overall quality of the paper suitable?

Excellent

Is the length of the paper justified?

Yes

Should the paper be seen by a specialist statistical reviewer?

No

Do you have any concerns about statistical analyses in this paper? If so, please specify them explicitly in your report.

No

It is a condition of publication that authors make their supporting data, code and materials available - either as supplementary material or hosted in an external repository. Please rate, if applicable, the supporting data on the following criteria.

Is it accessible?

N/A

Is it clear?

N/A

Is it adequate?

N/A

Do you have any ethical concerns with this paper?

No

Comments to the Author

This paper reviews the literature on gene drive systems that have not evolved suppressors. Some gene drive systems persist as low to intermediate stable frequencies over long periods of evolutionary time. This is very important for understanding the evolution of drive systems and interesting to consider for those designing synthetic gene drive systems. The authors do a nice job of setting up and then evaluating evidence for several possible explanations for apparently unsuppressed drive, focusing in particular on Sex Ratio in *Drosophila pseudoobscura*.

This is a thoughtful review and I really enjoyed reading it. I have some minor comments that I detail below:

Page 4: "Mullerian degredation"=="Muller's Ratchet"?

Page 4: The authors make a generalization about avoiding suppression and achieving fixation being critical for synthetic drive systems. However the latter (fixation) depends on the synthetic drive strategy one is employing. Fixation of the driver is not always the goal.

Table 1: The objective of the table was not clear to me. In particular, it is not clear what the criteria were for a drive system to make it into this table (drive without suppression?). For that reason, it wasn't clear why *D. persimilis* and *D. paramelanica* are included if the duration of time that the SR was observed was N/A and 1960s (the latter is confusing because Stalker's paper was published in 1961).

Page 7: "enormous population size". This is relative. Among *Drosophila* species, hundreds of thousands does not seem enormous. These two sentences left me confused about differences between census and effective population sizes in this species. I think a slight change of wording could make this clear.

Page 8: I think it is not unlikely that you would find a population with no (longer any) SR but with a suppressor, depending on the population dynamics of the system, costs of the suppressor etc. I believe this has been observed in autosomal segregation distortion systems (SD in *Drosophila melanogaster*).

Page 9: I'm not sure why it is that unlikely that a novel form of SR could emerge that is resistant to suppressors and that this could happen frequently enough that it would appear unsuppressed and stable over 80 years. Similarly on page 16: do you know that this is the same SR chromosome and not a new variant of this SR (perhaps with a different modifier)? If so, it would help to provide evidence here.

Page 14: Polyandry section: "In addition, the SR X chromosome in *D. pseudoobscura* occur" -> "occur" should be "occurs"

Decision letter (RSPB-2019-2267.R0)

29-Oct-2019

Dear Nina,

Your manuscript has now been peer reviewed and the reviews have been assessed by an Associate Editor. The reviewers' comments (not including confidential comments to the Editor)

and the comments from the Associate Editor are included at the end of this email for your reference. As you will see, the reviewers and the Editors have raised some concerns with your manuscript and we would like to invite you to revise your manuscript to address them.

Research ethics:

Use of animals and field studies:

Please submit a copy of your revised paper within three weeks. If we do not hear from you within this time your manuscript will be rejected. If you are unable to meet this deadline please let us know as soon as possible, as we may be able to grant a short extension.

Best wishes,
Professor Loeske Kruuk
Editor
mailto: proceedingsb@royalsociety.org

Associate Editor
Board Member: 1
Comments to Author:

This is a very readable paper which brings in fresh questions about the evolution of drive suppressors. The referees were generally positive about it but both have suggestions for revisions.

I'm not aware that Larner et al. 2019 is in press, and having handled that paper, see that some of the reviewer's criticisms of that paper would also seem to apply to the description of that study within this manuscript. I ask the authors to address this point as well.

Reviewer(s)' Comments to Author:

Referee: 1

Comments to the Author(s)

Price, Verspoor and Wedell present a thought-provoking discussion of factors that could lead to long-term persistence of meiotic drivers. I thank the authors for highlighting such an interesting

but understudied area of the meiotic drive field. I hope this work stimulates more analysis in this area which may further challenge the assumption that meiotic drivers are generally short-lived. I think the discussion in this work is sound, but I offer a few suggestions below intended to be helpful in improving the paper.

Major Comment:

Using the *D. pseudoobscura* SR system as the primary example is challenging because the driver(s) responsible for the phenotype is (are) unmapped. I think focusing on SR is fine, but I suggest that the authors lay out explicitly that it is unmapped when SR is introduced. In addition, the main arguments are based on the assumption that one genetic drive system underlies the observed SR phenotypes studied since the 1930s. I found it difficult to shake the possibility that there were multiple drivers while reading the paper because several species harbor multiple meiotic drive systems. The conflicting results regarding female fecundity could also be explained by multiple drivers. It would therefore be useful for the authors to lay out their arguments for why they assume that one driver, rather than several, underlies all the observations over time. For example, the authors cite a paper from 1965 regarding the mechanism of SR. Does the phenotype look similar in SR flies collected more recently? Importantly, I am not suggesting that a reader needs to be absolutely convinced about one driver underlying SR for the rest of the discussion to be stimulating/informative. I do appreciate that the authors are using SR as a model to illustrate ideas that could apply broadly to generic drive systems.

Minor Comments:

-Table 1: Why is *D. paramelanica* on this table? There is no range in dates provided. For *D. neotestacea*, why does the range go to 2014 when the last paper was published in 2013?

-p 10: The simplest drive systems do not require 2 loci. The Spok and wtf systems in fungi both work with one gene.

-Figure 1: Distinguishing the colors within the orange and blue groups is difficult. Perhaps use a more dramatically different color palette.

-Figure 2: The placement of the suppressors might give the impression that they must be at an allelic locus (e.g. on the Y chromosome). That is not true and would really limit the genetic real estate available to suppressors. That could lead a reader to be less impressed than they should be at the lack of observed suppressors. Then again, did any of the searches for suppressors look for autosomal suppressors of SR, or only resistant Y chromosomes?

-p 14: Is the Brandt reference a typo?

-p 17: I disagree that those are the only 'well understood' drive systems. There are systems in plants and fungi that are at least as well (sometimes better) understood than those in animals.

Referee: 2

Comments to the Author(s)

This paper reviews the literature on gene drive systems that have not evolved suppressors. Some gene drive systems persist at low to intermediate stable frequencies over long periods of evolutionary time. This is very important for understanding the evolution of drive systems and interesting to consider for those designing synthetic gene drive systems. The authors do a nice job of setting up and then evaluating evidence for several possible explanations for apparently unsuppressed drive, focusing in particular on Sex Ratio in *Drosophila pseudoobscura*.

This is a thoughtful review and I really enjoyed reading it. I have some minor comments that I detail below:

Page 4: "Mullerian degredation"=="Muller's Ratchet"?

Page 4: The authors make a generalization about avoiding suppression and achieving fixation being critical for synthetic drive systems. However the latter (fixation) depends on the synthetic drive strategy one is employing. Fixation of the driver is not always the goal.

Table 1: The objective of the table was not clear to me. In particular, it is not clear what the criteria were for a drive system to make it into this table (drive without suppression?). For that reason, it wasn't clear why *D. persimilis* and *D. paramelanica* are included if the duration of time that the SR was observed was N/A and 1960s (the latter is confusing because Stalker's paper was published in 1961).

Page 7: "enormous population size". This is relative. Among *Drosophila* species, hundreds of thousands does not seem enormous. These two sentences left me confused about differences between census and effective population sizes in this species. I think a slight change of wording could make this clear.

Page 8: I think it is not unlikely that you would find a population with no (longer any) SR but with a suppressor, depending on the population dynamics of the system, costs of the suppressor etc. I believe this has been observed in autosomal segregation distortion systems (SD in *Drosophila melanogaster*).

Page 9: I'm not sure why it is that unlikely that a novel form of SR could emerge that is resistant to suppressors and that this could happen frequently enough that it would appear unsuppressed and stable over 80 years. Similarly on page 16: do you know that this is the same SR chromosome and not a new variant of this SR (perhaps with a different modifier)? If so, it would help to provide evidence here.

Page 14: Polyandry section: "In addition, the SR X chromosome in *D. pseudoobscura* occur" -> "occur" should be "occurs"

Author's Response to Decision Letter for (RSPB-2019-2267.R0)

See Appendix A.

Decision letter (RSPB-2019-2267.R1)

19-Nov-2019

Dear Dr Wedell

I am pleased to inform you that your manuscript entitled "Ancient gene drives: an evolutionary paradox" has been accepted for publication in Proceedings B.

Open Access

Paper charges

Sincerely,

Professor Loeske Kruuk
Editor, Proceedings B
<mailto:proceedingsb@royalsociety.org>

Associate Editor:

Board Member

Comments to Author:

(There are no comments.)

Appendix A

Dear Editor,

Please find attached a letter outlining how we have addressed the comments raised by the referees. The main issue that was raised by both referees, concern the potential issue of multiple drivers and suppressors in *D. pseudoobscura*. We have carefully considered this possibility and have amended the MS to reflect this. However, the data we have to date do not support this suggestion. Below we outline how we have addressed each of the points raised in turn.

Associate Editor
Board Member: 1
Comments to Author:

This is a very readable paper which brings in fresh questions about the evolution of drive suppressors. The referees were generally positive about it but both have suggestions for revisions.

I'm not aware that Lerner et al. 2019 is in press, and having handled that paper, see that some of the reviewer's criticisms of that paper would also seem to apply to the description of that study within this manuscript. I ask the authors to address this point as well.

This likely stem from our own confusion as to how to cite submitted papers under review in this MS. We were led to believe that we could refer to them by using 'author 2019', but clearly this should only apply to accepted MS. Our mistake - apologies. However, we are happy to say that the MS has now been accepted.

Referee: 1

Major Comment:

*Using the *D. pseudoobscura* SR system as the primary example is challenging because the driver(s) responsible for the phenotype is (are) unmapped. I think focusing on SR is fine, but I suggest that the authors lay out explicitly that it is unmapped when SR is introduced. In addition, the main arguments are based on the assumption that one genetic drive system underlies the observed SR phenotypes studied since the 1930s. I found it difficult to shake the possibility that there were multiple drivers while reading the paper because several species harbor multiple meiotic drive systems. The conflicting results regarding female fecundity could also be explained by multiple drivers. It would therefore be useful for the authors to lay out their arguments for why they assume that one driver, rather than several, underlies all the observations over time. For example, the authors cite a paper from 1965 regarding the mechanism of SR. Does the phenotype look similar in SR flies collected more recently? Importantly, I am not suggesting that a reader needs to be absolutely convinced about one driver underlying SR for the rest of the discussion to be stimulating/informative. I do appreciate that the authors are using SR as a model to illustrate ideas that could apply broadly to generic drive systems.*

Yes we agree that fact we do not know the molecular basis to SR in *D. pseudoobscura* means there are some obvious limitations, and we have now stated up front the pros and con of focussing on the SR system (lines 143-146). We explicitly state that SR is unmapped. But on the plus side, SR is well studied in the lab and the field, so we know a lot about the fitness consequences to both sexes and in turn how this may scale up to affect the population dynamics of SR in natural populations, which is the focus of our review.

The question of whether the modern SR driver is substantially different to that originally studied in the 30s is an interesting one. There is no conclusive evidence, because the population genetics of SR, and particularly historical SR, has been surprisingly little studied. However, we think a major change in SR is unlikely.

Firstly, all SR chromosomes have shown consistent phenotypes, that do not differ substantially from any of the reports since the 1940s. None have ever produced fertile sons. In terms of karyotype, all reported SR chromosomes to date have the two large inversions on the X right arm. Beckenbach 1996 found a single driving SR chromosome that lacked the small terminal inversion, but still showed full drive. Fuller et al 2018 found recombined X chromosomes that had the small terminal SR inversion, but did not show any drive. So the karyotype for SR has been consistent since at least the 1940s.

While replacement of one driver by another variant is possible, we think this is unlikely. A sweep to completely replace SR across the USA would need a major fitness improvement in the sweeping SR variant. Two obvious potential causes of this are:

- 1) Replacement of the original SR by a new version that has higher fitness due to a stronger drive (i.e. a drive variant that gets into 90% of sperm outcompetes an original variant getting into 80% of functional sperm). But SR drive has been 100% effective since the 1940s. So selection on drive strength cannot be causing replacement.
- 2) It could potentially be due to constant conflict between drive and suppression- i.e. drive strength is reduced due to the evolution of suppression, and a higher drive variant escapes suppression and replaces the original (see point 1). But this would predict a mosaic of drive variants and suppressors, at

least transiently, as has been seen in both *D. simulans* and *D. paramelanica*. But no evidence for anything like this has been found in *D. pseudoobscura* despite extensive field sampling.

You could also get replacement by something much less likely to create major fitness differences:

- 3) You could also get drive replacement via selection on SR for lower costs, which might explain some of the differences in results between groups researching the costs of SR. But it is not clear that this would lead to a sweep unless SR carries a major cost that is suddenly removed, which would be needed to create a strong selective advantage. If improvements create marginal fitness benefits, or fitness benefits specific to a situation or local environment, again you would not get a major sweep.

Moreover, if a higher fitness variant did sweep across the USA for ANY of the above reasons, it would be expected to alter the overall fitness of SR relative to ST. So we should have evidence of changes in frequencies of SR in populations. But instead, we see remarkable consistency in the frequency of drive across the species range sampled over 70+ years.

So the most likely scenario is that drive is under consistent evolutionary pressures, and is unlikely to have radically changed. We have now added a more explicit discussion of this (lines 196-202).

We now also better address the reasons for different findings regarding the fitness cost to SR carrying females and conclude it most likely can be explained by a combination of different methodologies and uncontrolled confounding variables (lines 413-417)

Minor Comments:

-Table 1: Why is D. paramelanica on this table? There is no range in dates provided. For D. neotestacea, why does the range go to 2014 when the last paper was published in 2013?

You are correct about both. For *D. paramelanica*, we only have one distribution report, so we have changed the range to "N/A". For *D. neotestacea* we simply typed the date incorrectly. We apologise for both errors.

-p 10: The simplest drive systems do not require 2 loci. The Spok and wtf systems in fungi both work with one gene.

We have now amended the text accordingly (lines 242-243).

-Figure 1: Distinguishing the colors within the orange and blue groups is difficult. Perhaps use a more dramatically different color palette.

We have now tried a range of colour palettes for this figure. Unfortunately, they all looked much worse (we tested it on several of our colleagues), and we could not find any that were better than the orange and blue we originally used.

-Figure 2: The placement of the suppressors might give the impression that they must be at an allelic locus (e.g. on the Y chromosome). That is not true and would really limit the genetic real estate available to suppressors. That could lead a reader to be less impressed than they should be at the lack of observed suppressors. Then again, did any of the searches for suppressors look for autosomal suppressors of SR, or only resistant Y chromosomes?

We have now adjusted figure 2 to show the possibility of suppression evolving on non-homologous chromosomes. We thank the reviewer for bringing our attention to this detail and it was not our intention to exclude the possibility of autosomal suppression.

The methods used to find suppression of SR tested both autosomes and Y chromosomes (briefly detailed in Beckenbach et al 1982, now added to the reference list for the MS). In brief, SR was crossed in to a wide variety of lines of flies, creating males carrying SR plus a random wild Y chromosome, with half their autosomes derived from the wild line. This should test for both autosomal and Y suppression. These crosses were made using flies collected from California to Texas. The only possibility that has not been extensively tested (to our knowledge) would be recessive autosomal suppression, but this seems both unlikely to evolve and of questionable evolutionary consequence.

Beckenbach A, Curtsinger JW, Policansky D. 1982. Fruitless experiments with fruit flies: the "sex ratio" chromosomes of *D. pseudoobscura*. *Drosophila Information Service* 58: 22.

-p 14: Is the Brandt reference a typo?

Yes, now corrected.

-p 17: I disagree that those are the only 'well understood' drive systems. There are systems in plants and fungi that are at least as well (sometimes better) understood than those in animals.

We have now toned down this statement and acknowledge the limitations with SR in comparison with other more well characterised drive system from the molecular perspective and emphasise we are specifically referring to the ecology and natural history of SR and that it has been one of the longest study meiotic drive systems (lines 510-511).

Referee: 2

This is a thoughtful review and I really enjoyed reading it. I have some minor comments that I detail below:

Page 4: "Mullerian degredation"="Muller's Ratchet"?

Now changed.

Page 4: The authors make a generalization about avoiding suppression and achieving fixation being critical for synthetic drive systems. However the latter (fixation) depends on the synthetic drive strategy one is employing. Fixation of the driver is not always the goal.

This is a good point, and we have now moderated this statement to make this distinction clear (lines 65-67).

Table 1: The objective of the table was not clear to me. In particular, it is not clear what the criteria were for a drive system to make it into this table (drive without suppression?). For that reason, it wasn't clear why *D. persimilis* and *D. paramelanica* are included if the duration of time that the SR was observed was N/A and 1960s (the latter is confusing because Stalker's paper was published in 1961).

Table 1 is aimed at first showing that there is geographic variation in drive frequency in several species, not just *D. pseudoobscura*. The second aim is to show that for some of these species there is pretty convincing evidence that the drive frequency has been stable for a surprisingly long time. We have changed the legend to better explain this: "Examples of five *Drosophila* species that show geographic variation in the frequency of their X chromosome drive systems. The duration column shows the duration between the first observation of the distribution to the most recent work verifying that the distribution has not changed, for species where multiple surveys have been published."

Page 7: "enormous population size". This is relative. Among *Drosophila* species, hundreds of thousands does not seem enormous. These two sentences left me confused about differences between census and effective population sizes in this species. I think a slight change of wording could make this clear.

This statement is now reworded.

Page 8: I think it is not unlikely that you would find a population with no (longer any) SR but with a suppressor, depending on the population dynamics of the system, costs of the suppressor etc. I believe this has been observed in autosomal segregation distortion systems (SD in *Drosophila melanogaster*).

Yes in theory this could occur, but we think unlikely to remain for extended period of time in the absence of drive for the reasons highlighted by referee 1. Importantly, drive suppressors most likely incur costs. However as outlined above, we have now modified the text when discussing the evidence and likelihood of drive and suppressor dynamic in the SR system.

Page 9: I'm not sure why it is that unlikely that a novel form of SR could emerge that is resistant to suppressors and that this could happen frequently enough that it would appear unsuppressed and stable over 80 years. Similarly on page 16: do you know that this is the same SR chromosome and not a new variant of this SR (perhaps with a different modifier)? If so, it would help to provide evidence here.

This is an interesting point which the other reviewer also suggested. Please see our response to that point above. In short, we cannot be 100% sure it is the same driver that is observed across the 80 years of surveying. However, the karyotype and phenotype have been very consistent since the 1940s. There has been no evidence

of a mosaic of suppression and effective/ineffective SR variants, as has been seen in *D. simulans* and *D. paramelanica* despite reasonably extensive sampling. Moreover, any selection powerful enough to replace an SR variant is likely to also alter the balance of fitness between SR and ST, and so is likely to be seen in changes in SR frequency in populations. As outlined in the response to referee 1 above, we have now provided more information about what is known about SR in *D. pseudoobscura*.

But ultimately, we need to wait until someone has done detailed population genetics on SR to really answer this question. Hopefully Nitin Phadnis' team are working on this.

Page 14: Polyandry section: "In addition, the SR X chromosome in D. pseudoobscura occur"—> "occur" should be "occurs"

Corrected.

We hope that these corrections are sufficient for a final acceptance of the MS.

Nina Wedell, on behalf of all the authors.